# Chimera and Tandem-Repeat Type Galectins: The New Targets for Cancer Immunotherapy

**DOI:** 10.3390/biom13060902

**Published:** 2023-05-29

**Authors:** Frankie Chi Fat Ko, Sheng Yan, Ka Wai Lee, Sze Kwan Lam, James Chung Man Ho

**Affiliations:** 1Department of Medicine, School of Clinical Medicine, Li Ka Shing Faculty of Medicine, The University of Hong Kong, Queen Mary Hospital, 102 Pokfulam Road, Hong Kong, China; ssyan@hku.hk (S.Y.);; 2Pathology Department, Baptist Hospital, Waterloo Road, Kowloon, Hong Kong, China

**Keywords:** galectins, intracellular, secretory, immune checkpoint inhibitors, immuno-oncology therapy

## Abstract

In humans, a total of 12 galectins have been identified. Their intracellular and extracellular biological functions are explored and discussed in this review. These galectins play important roles in controlling immune responses within the tumour microenvironment (TME) and the infiltration of immune cells, including different subsets of T cells, macrophages, and neutrophils, to fight against cancer cells. However, these infiltrating cells also have repair roles and are hijacked by cancer cells for pro-tumorigenic activities. Upon a better understanding of the immunomodulating functions of galectin-3 and -9, their inhibitors, namely, GB1211 and LYT-200, have been selected as candidates for clinical trials. The use of these galectin inhibitors as combined treatments with current immune checkpoint inhibitors (ICIs) is also undergoing clinical trial investigations. Through their network of binding partners, inhibition of galectin have broad downstream effects acting on CD8^+^ cytotoxic T cells, regulatory T cells (Tregs), Natural Killer (NK) cells, and macrophages as well as playing pro-inflammatory roles, inhibiting T-cell exhaustion to support the fight against cancer cells. Other galectin members are also included in this review to provide insight into potential candidates for future treatment(s). The pitfalls and limitations of using galectins and their inhibitors are also discussed to cognise their clinical application.

## 1. Introduction

The function of the immune system in fighting cancer cells has been of long-standing interest in the context of cancer therapies [1], with research dating back to William B. Coley’s study in 1819 [2]. Since the first ICI, Cytotoxic-T-lympocyte-antigen-4 (CTLA-4), also known as ipilimumab (Yervoy), was tested and approved for the treatment of metastatic melanoma in 2015 [3,4], the number of checkpoint inhibitors has increased. In particular, when programmed cell death/ligand-1 (PD-1/PD-L1) immune checkpoint proteins are identified, these checkpoint inhibitors are now the standard regimen for immuno-oncology (I-O) therapy when tackling different solid tumours [5]. The most common PD-1 and PD-L1 inhibitors are pembrolizumab (Keytruda), nivolumab (Opdivo), cemiplimab (Libtayo), atezolizumab (Tecentriq), nivolumab (Bavencio), and durvalumab (Imfinzi) [6]. In addition, ipilimumab (Yervoy) is the most common CTLA-4 inhibitor. However, the response rates (RRs) of this approach vary according to types and lines of treatment [7]. It has been found to have a good-to-moderate response, with over a 50% RR against classic Hodgkin’s lymphoma, melanoma, and first-line combination-treated non-small cell lung cancer (NSCLC). However, some cancers, such as extensive-stage small cell lung cancer (SCLC), hepatocellular carcinoma (HCC), PD-L1^+^ gastric (gastroesophageal junction type), and cervical cancers, have shown less than a 25% RR.

The current challenges to the efficiency of I-O therapies include the exhaustion of cytotoxic T cells [8] and the need to increase the subpopulation of Tregs and other immune cells during immunosuppression [9]. To combat these challenges, other immunomodulators have been identified, such as T-cell immunoreceptor with immunoglobulin and ITIM domains (TIGIT). TIGIT is found in T cells, NK cells, and tumour cells. The mechanism of immune inactivation may occur through ITIM-dependent negative pathways [10]. Numerous TIGIT antibodies, including BMS-986207, tiragolumab, and vibostolimab, have been developed and tested in clinical trials [11]. The recent results of these trials, such as CITYSCRAPE [12], have generated additional interest in investigating any novel immunosuppressors and their interacting protein mechanisms. The results of the subsequent phase III trials, such as the SKYSCRAPER series (NCT04619707, NCT04513925, NCT046665843, etc.), will need further analysis to enhance this approach to treatment. A list of this series of studies was summarised by Brazel et al. (2023) [13].

Recently, galectins have been identified as immunomodulators [14], joining TIGIT as new potential targets for immunotherapies. The potential candidate reagents and ongoing trial studies are listed in Table 1. In particular, newly developed reagents, such as GB1211 [15], a galectin-3 small-molecule inhibitor, and LYT-200, an anti-galectin-9 humanised antibody, are currently on trial (NCT05240131/GALLANT-1 and NCT04666688, respectively). The safety of GB1211 has also been reported, with limited grade 1 and grade 2 adverse effects in healthy participants (NCT03809052) [16]. The efficiency of these candidates in the current trial studies will further support the use of galectins in cancer treatments in combination with PD-1/PD-L1 and TIGIT. Other galectin inhibitors, such as OTX008 and ProLectin-M, have also been investigated in mouse animal model studies [17,18,19,20]. Other newly identified small-molecule/peptide inhibitors and antibodies also represent new potential agents for treatment and are discussed in Section 3.1 below.

## 2. Human Galectins

There are 12 galectins in humans, as listed in Table 2. The genomic locations of these genes and their protein product structures are shown in Figure 1 and Figure 2, respectively. Galectins can be detected in the cytoplasm and exhibit a secretory form. Protein topology analysis [24] has shown that conventional secretory proteins, which contain an N-terminal signal peptide that starts with a few positively charged amino acids, such as lysine (K) and arginine (I), followed by around 12–16 hydrophobic amino acids [25], are driven into the endoplasmic reticulum (ER). The secretory proteins are then embedded inside lipid bilayer vesicles and transported via budding-off into the Golgi apparatus, after which the vesicles can be further fused with the cell membrane to export the protein outside the cell [26]. However, further analysis of the coding and protein sequence of galectins has resulted in no signal peptide sequence being detected. Their secretory forms are suspected to be produced through the non-canonical secretion pathway [27,28]. Unlike other proteins, their recognition is not based on protein peptides in the form of amino acid chains on the binding partner(s). Galectins contain a carbohydrate-recognition domain (CRD) as a binding motif to recognise the glycosylation sites on other proteins and for binding. The CRD mainly detects glycoproteins and is required for post-translational modification. Their immunological roles have been established and reviewed [29]. Different galectins have been identified that bind with immune cells, such as cytotoxic T cells, dendritic cells, and macrophages, to regulate cancer cell immunosurveillance (also listed in Table 2). The roles of these human galectins are discussed in further detail below.

### 2.1. Galectin-3: The Only Chimera-Type Galectin with Oncogenic Functioning

Among all the members of the galectin family, only galectin-3 contains a 12-amino-acid N-terminus with a serine (S) residue at position six for phosphorylation, followed by a 100-amino-acid collagen-like sequence (CLS) domain and a 130-amino-acid CRD at the C-terminus [30] as a single protein unit of about 28 kDa in size. The N-terminus of galectin-3 facilitates the formation of a multi-unit chimera complex, as shown in Figure 2. Post-translational modifications, such as phosphorylation on serine 6, can promote tumour aggressiveness and metastasis, whereas mutating this residue into alanine (A) caused the tumorigenicity of breast cancer cells implanted in nude mice to be abolished [31]. In terms of its intracellular roles, galectin-3 has been found to bind different interacting partners to control tumorigenicity. It can bind with K-Ras [32] to activate downstream signalling pathways, including PI3K/Akt, PLC/PKC, Raf/MEK/ERK, RALGDS/Ral, and TIAM1/Rac, to promote cancer cell survival, migration, invasion, and cell cycle progression through the hijacking of the upstream receptor tyrosine kinases, such as epidermal growth factor receptor (EGFR) and insulin-like growth factor receptors (IGFRs), as well as G-protein coupling receptors (GPCRs) [33]. In addition, it can also interact with the proline-rich region (PRR) of ALG-2-interacting protein X (Alix) for an endosomal-sorting complex functioning [34,35]. Alix-depleted basal-like breast cancer cells have also been shown to induce EGFR activity and EGFR-dependent PD-L1 presentation [36]. In addition to these cytoplastic roles, galectin-3 can also interact with the β-catenin signalling component PCDH24 and spliceosome complex member GEMIN4 to regulate gene expression and alter isoform expression [37,38]. Galectin-3 has also been found to upregulate p21 expression via Sp1 binding to control the cell cycle progression [39].

Interestingly, galectin-3 also plays an extracellular role in controlling cancer growth. In NSCLC, inhibition of galectin-3 increases the oxidative stress in vitro (in A549 and H1792 cell lines) and PDX models [40]. Galectin-3 acts on the integrin αvβ3 extracellularly via K-Ras to promote oncogenesis. In contrast, overexpression of β3 in the null cell line H727 promoted the growth of cancer cells, which were perturbed by shRNA KRAS or galectin-3 knockdown.

In addition to galectin-3 promoting pro-oncogenic roles in cancer cells, it also acts on other immune cells for immunosuppression. In healthy individuals, galectin-3 plays an important role in systemic innate immunity for pathogen recognition as a pattern-recognition receptor (PRR)/danger-associated molecular pattern (DAMP) and is secreted by myeloid cells, including neutrophils, monocytes, macrophages, and dendritic cells (DCs) [41]. According to binding assays of galectin-3 and chemokines, galectin-3 can bind to numerous chemokines, including CCL1, -5, -13, -19, -20, -21, -22, -24, -25, -26, -27, and -28 and CXCL7, -10, -12, -13, -14, -16, and -17 to perturb the migration properties of different immune cells [42]. Galectin-3 also plays a direct role in killing effector T cells through CD71 and integrin-associated protein (IAP) binding [43]. It can also inhibit macrophage cytotoxicity and phagocytosis [44]. Cancer cells secrete galectin-3 within the TME to serve in a general immunosuppressive role.

Numerous galectin-3 inhibitors have been identified that tackle the immunomodulating role of immune cells. Belpectin (GR-MD-02) was developed to prevent non-alcoholic steatohepatitis (NASH) cirrhosis [45], which is a risk factor for developing HCC [46]. GB1211 is the first oral bioavailable galectin-3 inhibitor with an affinity to human galectin-3 at K_d_ = 0.025 μM. It was modified from the inhaled galectin-3 inhibitor GB0139 for the treatment of idiopathic pulmonary fibrosis [15] (Figure 2). Belpectin and GB0139 were developed to inhibit fibrosis via the control of galectin-3 [47]. A stage I clinical trial study saw pembrolizumab combined with belpectin against metastatic melanoma or head and neck squamous cell carcinoma (HNSCC) (NCT02575404, listed in Table 1), and the safety and efficacy were assessed [21]. The results were promising; the combination reactivated effective memory T cells and reduced suppressive monocytic myeloid cells, showing mainly grade I adverse effects (with no grade III and IV). Another galectin-3 inhibitor, GB1211, has been combined with atezolizumab for use against NSCLC (stage I/II) and is undergoing trials. The results and their extension to phases II and III will provide further insight into the applicability of galectin-3 as a combinatorial treatment with standard ICI (PD-1/PD-L1).

**Table 2 biomolecules-13-00902-t002:** Intracellular and extracellular binding partners of 12 human galectins.

Gene/Protein Name (Chromosome Position [48])	Intracellular (*Cytoplasmic*/*Nucleus*)	Extracellular
LGALS1/Galectin-1 (Chr. 22q13.1)	*Cytoplasmic*: GRP78 [49] Gemin4 [37] H-Ras [50] PCDH24 [38]	CC and CXC chemokines [42] CD43 [51,52] CD45 [51,53] NRP1 [54] VEGFR2 [55]
LGALS2/Galectin-2 (Chr. 22q13.1)		Binds to surface of CD14^(interm.–high)^ monocyte and promote M1 macrophage differentiation [56]
LGALS3/Galectin-3 (Chr. 14q22.3)	*Cytoplasmic*: Alix (EGFR trafficking) [34,35,57] Gemin4 [37] K-Ras [32,40] PCDH24 [38] *Nucleus*: hnRNPA2B1 [58] Sp1 [39]	CC and CXC chemokines [42] CD29 [43] CD43 [43] CD45 [43] CD71 [43] EGFR [59] Interferon-γ [60] Integrin α_v_β_3_ [40] LAG3 [61] MUC1 [62]
LGALS4/Galectin-4 (Chr. 19q13.2)		CD3 [63]
LGALS7/Galectin-7 (Chr. 19q13.2)	*Cytoplasmic*: Bcl-2 [64]	
LGALS8/Galectin-8 (Chr. 1q43)		αM (CD11b, neutrophils) [65] CD166 [66] Podoplanin [67,68]
LGALS9/Galectin-9 (Chr. 17q11.2)	*Cytoplasmic*: Binding to intracellular TIM-3 to modulate mTOR phosphorylation [69] *Cytoplasmic–Lysosomes*: Interact with Lamp2 to regulate lysosomal functions and autophagy [70]	4-1BB [71] CD40 [72] CD44 [73] CD206 [74] Dectin-1 (macrophages) [75] DR3 [76] PD-1 [77] PDI [78,79] TCR [80,81] TIM-3 [77,82,83] VISTA [84]
LGALS10/Galectin-10/Charcot-Leyden crystal protein CLC (Chr. 19q13.2)	*Cytoplasmic–Granules*: Eosinophil-derived neurotoxin EDN (RNS2) and eosinophil cationic protein ECP (RNS3) co-localised with CD63. It is required for the maturation of eosinophil during granulogenesis [85]	
LGALS12/Galectin-12/GRIP1 (Chr. 11q12.3)	*Cytoplasmic–Endosome/Lysosomes*: VPS13C in lipid droplets and promotes the polarisation to M1 macrophage via TLR4 pathway [86,87]	
LGALS13/Galectin-13/ placental protein 13 (Chr. 19q13.2)	*Nucleus*: HOXA1 [88]	Binds to T lymphocytes and induces apoptosis [89]; Binds to neutrophils and shifts to immunoregulatory phenotype and promotes high PD-L1 expression [90]
LGALS14/Galectin-14 (Chr. 19q13.2)		Binds to T lymphocytes and induces apoptosis [89] c-Rel [91]
LGALS16/Galectin-16 (Chr. 19q13.2)		c-Rel [92]

Remarks: Colour code is based on the galectin’s structure: chimera type (galectin-3) is highlighted in light blue; tandem-repeat type is highlighted in light green; prototype has not been highlighted.

### 2.2. Tandem-Repeat Type Galectins

#### 2.2.1. Galectin-9 Acts on the Immunosuppression of Cancers

Galectin-9 was first identified in 1997 in embryonic mouse kidneys as a 36 kDa protein [93] with two distinct CRDs (as shown in Figure 2) and is associated with human Hodgkin’s lymphoma [94]. It has been shown to be overexpressed in different cancers (Table 3). High mobility group box 1 (HMGB1) has been demonstrated to induce the Toll-like receptor 4-mediated pathway to promote galectin-9 expression in cancer cells [95]. However, associations between the overexpression of galectin-9 in tumour tissues (immunohistochemistry (IHC) staining) and clinicopathological/survival parameters have been quite controversial in different studies [96,97]. The galectin-9 that is produced by cancer cells seems to work dynamically with the surrounding cells. In pancreatic adenocarcinoma, the expression of galectin-9 is positively correlated with PD-L1 in tumour tissues [97]. Interestingly, the serum level of galectin-9 is somehow reflective of the staging of colon cancer and NSCLC patients [98]. Galectin-9 may be a good diagnostic and prognostic marker for cancer staging. However, the role of systemic serum galectin-9 is still elusive. In contrast, galectin-9 overexpression in isolated tumour-infiltrating lymphocytes (TILs) at a tumour site represents T-cell exhaustion and impairment in NK cells [99]. Hence, the roles of infiltrating and peripheral leukocytes may be different within the TME and systemic immunity, respectively.

The interacting partners of galectin-9 have been identified and are mainly expressed on the cell surface of different immune cells, including NK cells, effector T cells, Tregs, monocytes, macrophages, and neutrophils (Table 2 and Figure 3). Galectin-9 studies have revealed an immunoregulation [105] within the TME. These molecules are displayed as transmembrane proteins that transduce signals from the cell surface to intracellular effectors to control immune cell proliferation, survival, and cytokine secretion [106]. All these molecules contain either an immunoreceptor tyrosine-based activation motif (ITAM) or an inhibitory motif (ITIM) domain that further activates or inhibits downstream activities, respectively. ITAM-containing proteins recruit Src/Lck proteins for phosphorylation to activate downstream processes. In contrast, ITIM-containing proteins recruit phosphoinositide phosphatase (SHIP) proteins to dephosphorylate the downstream components via hydrolysation for the inactivation [107]. Galectin-9 has been identified to bind to PD-1 and TIM-3 in CD8^+^ cytotoxic T cells [77,82]. In addition, it was also shown to interact with T-cell receptor (TCR) on Treg, which has also been found to suppress cytotoxic T via the galectin-9–TIM-3 axis in HIV contexts [108]. However, in T-cell/tumour-cell co-culture environments, galectin-9 induces T-cell apoptosis, which can be blocked by galectin-9 antibodies [80]. The dominant role of galectin-9 in the TME may still be pro-tumorigenic. TIM-3 is also an ITIM-containing protein that suppresses the proliferation of effector T cells, NK cells, and monocytes, which is similar to the roles of TIGIT, where TIGIT acts mainly on effector-T and NK cells (Figure 3) [109]. Other ITIM molecules, including CD40, VISTA, and 4-1BB, are activated by galectin-9 to suppress the expansion of effector T cells [71,72,73,84]. In addition, galectin-9 acts on VISTA to also suppress human cytotoxic T lymphocyte activity [84].

Galectin-9 also acts on Treg cell differentiation and stability through CD44 and DR3 to inhibit pro-inflammatory T helper 1 (Th1) and Th17 cells [110,111]. In addition, galectin-9 also acts on Dectin-1 for M2 macrophage polarisation in protective responses and on protein VEGF for angiogenesis as pro-oncogenic roles in the cancer cells [74,75]. In a more complicated situation, galectin-9 acts on both modulators for activation CD44 and inhibition TIM-3 on NK cells in mice [112]. However, in human NK cells, Gal-9^+^ NK cells only promoted IFNγ expression and not the expression of cytolytic molecules when compared to their control counterpart. The controlling role of galectin-9 in NK cell functioning may be different between humans and mice.

Inhibiting galectin-9 may help to activate adaptive immunity during cancer treatment, regarding the ICIs PD-1/PD-L1 [73,76]. Due to the limited role of galectin-9 intracellularly, instead of small-molecule inhibitors, humanised monoclonal antibodies can also serve as inhibitory reagents. LYT-200 is one of the fully humanised IgG_4_ monoclonal antibodies against galectin-9. A clinical trial study (NCT04666688) is currently ongoing, assessing the safety, pharmacokinetics, and efficacy of LYT-200 as a monotherapy or combined with tislelizumab (PD-1 inhibitor) or gemcitabine/nab-paclitaxel chemotherapy in patients with metastatic head and neck, colorectal, pancreatic, or urothelial cancers (see Table 1). The results might provide further insight into the role of galectin-9 inhibition in cancer treatments. In addition, a recent study suggests that circulating galectin-9 and PD-L1 levels are independent of their tumoural expression levels. Further detailed analyses of galectin levels should be conducted [113].

In addition to lymphocytes, neutrophils are the dominant leukocytes in the innate immune response through phagocytosis and NETosis [114,115]. An in vitro study demonstrated that galectin-9 could activate neutrophils to enact a killing effect on cancer cells [116], and neutrophils also contribute to the secretion of galectin-9 [117]. However, in clinical trials, neutrophil tumour infiltration and the neutrophil-to-lymphocyte ratio (NLR) have been found to be correlated with poor prognostic markers in patients with different solid cancers [118,119,120].

Recently, mucosal-associated invariant T (MAIT) cells were suggested to play roles in immunotherapy in both blood and solid tumours [121]. Galectin-9 has also been shown to deplete MAIT cells in CLL [122]. It will be interesting to see future research on the regulation of MAIT cells and the damping of immunotherapy by galectin-9.

#### 2.2.2. Other Tandem-Repeat Type Galectins: Galectin-4, -8, and -12

Galectin-4, -8, and -12 have been less well studied than galectin-1, -3, and -9. Although there is increasing interest in studying these molecules, we are still far from obtaining a complete picture. Galectin-4 was first identified as a ligand on a human blood group antigen [123]. Its expression has been detected in metastatic prostate cancer cells (Table 3) [124]. A high level of galectin-4 has also been shown to be associated with tissues in the advanced stages of breast and colorectal cancer [125,126]. Interestingly, galectin-8 has been found in malignant tumour tissues, especially in breast cancer [127]. Within the TME, secreted galectin-8 binds to podoplanin on tumour-associated macrophages and lymphatic endothelial cells to promote lymphangiogenesis and lymph node metastasis in breast cancer mouse models [67,68]. In contrast to most galectins, galectin-12 has been shown to promote the M1 macrophage polarisation [86,87] and may, therefore, play an opposite role(s) to other galectins in tumour suppression. The regulatory roles and inhibitory mechanisms of these three galectins in cancer development still need to be further investigated.

### 2.3. Prototype Galectins

#### 2.3.1. Galectin-1 Is a Prototype Galectin with Diverse Roles in Intra- and Extracellular Processes

Galectin-1 is an important protein in various biological processes, such as cell adhesion, migration, and cell death. Numerous studies have shown that galectin-1 is overexpressed in different tumours, including breast, lung, ovarian, pancreas, and prostate cancers (listed in Table 3). Intracellularly, galectin-1 binds to oncogenic H-Ras protein and facilitates H-Ras-to-membrane anchorage [50] and subsequent interaction with Raf kinase to further activate downstream oncogenic pathways [128]. In addition, galectin-1 also interacts with gem nuclear organelle association protein 4 (Gemin4), as does galectin-3 [37]. Gemin4 is important for microRNA biosynthesis in gene silencing [129]. Interestingly, polymorphism on Gemin4 is associated with poor prognosis in early-stage NSCLC patients [130]. Although the role of microRNAs in cancer cell gene regulation has become widely acknowledged, the mechanisms and roles still need to be elucidated. Another binding partner of galectin-1 is glucose-regulated protein 78 (GRP78) (Table 2). The overexpression of galectin-1 is also correlated with lymph node and distant metastasis in gastric cancer and advanced staging [49]. High expressions of galectin-1 and GRP78 also reflect poor overall survival in gastric cancer patients. In NSCLC cells, GRP78 is responsive to oxidative stress and promotes autophagy and epithelial–mesenchymal transition (EMT) [131]. Additionally, another galectin-1-interacting molecule, protocadherin LKC (PCDH24) [38], is also involved in the β-catenin pathway in EMT [132].

In the extracellular environment, the secretion of galectin-1 occurs through a non-canonical pathway, as with other members of the galectin family. It has been suggested that it is secreted through stress-induced exocytosis or the exosome pathway [27]. Similarly to galectin-3, secreted galectin-1 also binds to numerous chemokines, including CCL1, -5, -13, -20, -21, -22, -24, -25, -26, and -28 and CXCL9, -11, -12, -13, -16, and -17 for modulating the migration properties of different immune cells [42]. In addition, secreted galectin-1 binds to cell surface CD43 and CD45 to induce effector T-cell apoptosis for immunosuppression in the TME [52,133]. On the other hand, binding to CD43 on neutrophils was shown to promote neutrophil recruitment to the tumour site [53]. Galectin-1 can also bind to CD43/45 on monocyte-derived dendritic cells (MoDCs) for activation and migration [51]. MoDCs can further provide tolerogenic effects and suppress effector T cells through the PD-1/PD-L1 pathways [134]. The recruitment of dendritic cells and neutrophils can promote the Th17 polarisation [135] and cancer-cell priming [136] to promote the metastasis [136,137].

Galectin-1 also acts on endothelial cells. It has also been found to bind to vascular endothelial growth factor 2 (VEGF2) to control endothelial cell survival via the Akt pathway and promote angiogenesis around the tumour site [55]. Galectin-1 has also been found to bind to Neurophilin-1 (NRP-1) [54], which serves as the VEGFR coreceptors in endothelial cells for tumour angiogenesis [138].

On the clinical side, galectin-1 has been used as a prognostic marker for the prediction of treatment outcomes, even though the mechanism by which systemic/serum galectin-1 level increases is unknown. For example, in melanoma patients receiving treatment with the monoclonal antibody bevacizumab against VEGFR, a higher level of serum galectin-1 was associated with better overall survival following treatment [139]. On the other hand, patients who showed the overexpression of galectin-1 in their tumour tissue had poor overall survival in the advanced-stage small-cell lung cancer [140]. In colorectal cancer patients, galectin-1 attenuated cytotoxic CD8+ T-cell-killing activity and showed high expression, especially in stroma cells, which led to poor overall survival in colorectal cancer patients [141].

When galectin-1 has been used as a treatment or in clinical trials, it has been less promising than the other candidates (galectin-3 and galectin-9) mentioned above. Most of the studies are still on the bench side. OTX008 is the most commonly used galectin-1 inhibitor [142]. OTX008 has been examined in thyroid cancer cell lines, oral squamous cell carcinoma, hepatocellular orthotopic implantation models, and SCLC patient-derived xenograft-implanted nude mice. The tumour cells/implanted tumours treated with galectin-1 inhibitor OTX008 were significantly suppressed [18,20,140,143]. The limited availability of galectin-1 inhibitors may harm possible future clinical trial studies; further increasing the available specific inhibitors is discussed below and by Marino et al. [14].

#### 2.3.2. Another Prototype, Galectin-2, Has Controversial Roles in Different Cancers

Galectin-1 and galectin-2 share similar protein sequences and structures; however, their recognition and binding to carbohydrate side chains are different [144]. Extracellularly, secreted galectin-2 has been found to bind to CD14 on CD14^intermediate/high^ monocytes to promote M1 macrophage differentiation for a pro-inflammatory effect [56]. Based on our observations, the overexpression of galectin-2 can play a tumour-suppressive role in the H-Ras^G12V^-activated HCC cells [145], where H-Ras also plays an important role in HCC oncogenesis [146,147,148,149]. These findings may suggest that galectin-2 has a tumour-suppressive role. However, in triple-negative breast cancer, galectin-2 was found to act on immunosuppression in vivo to promote tumour growth but not to act on cancer cells [150]. These results suggest that galectin-2 may play dynamic regulatory roles in cancer and immune cell populations in terms of controlling tumour behaviour against different genetic backgrounds, such as H-Ras status. Increased serum galectin-2 was detected in circulating blood samples from breast and colon cancer patients and was correlated with different cytokines: IL-6; GCS-F; GROα/CXCL1; and MCP-1 [126]. It may serve as a prognostic marker. However, the systemic role of galectin-2 is still elusive.

#### 2.3.3. Other Prototype Galectins (-7, -10, -13, -14, and -16) Are Located on Chromosome 19

With the exception of galectin-4, which is classified as a tandem-repeat-type protein structure, as stated above, all the other galectin genes are clustered on chromosome 19 in the q13.2 position, including LGALS4, LGALS7, LGALS10, LGALS13, LGALS14, and LGALS16, and their protein products are classified as prototype galectins (Figure 1 and Figure 2). LGALS13, -16, and -14 are in a sense direction, whereas LGALS4, -7, and -10 are in an anti-sense direction [151]. LGALS7 and LGALS4 are upstream of the LGALS13, -16, and -14 clusters, whereas LGALS10 is downstream of these clusters.

Galectin-13 and -14, which are also known as placental galectins, can induce apoptosis in T lymphocytes. CD95 (Fas) and IL-8 could be induced in non-CD3-/CD28-activated T lymphocytes, particularly in the case of miscarriage [89]. The high expression of these galectins and placental galectin-16 is associated with poor prognosis and poor overall survival in breast and ovarian cancers [102]. These galectins are mainly expressed in the placenta and brain, with limited expression in the breasts, bone marrow, stomach, pancreas, retina, and skin. High percentages of thyroid and lung cancer tissues (>60%) express galectin-13 and galectin-14, respectively [102]. Greater attention should be paid to their immunological roles in cancer treatment. However, these placental galectins seem to act in the opposite direction in thyroid cancer. Intracellularly, galectin-13 can interact with the Hox family protein HOXA1 [88]. The upregulation of HOXA1 is also associated with poor prognosis in breast cancer and HCC patients [152,153]. Its intracellular role(s) should not be neglected. Galectin-14 can promote cell migration and invasion by upregulating MMP9 and N-cadherin in the trophoblasts [154]. Both galectins are also associated with pro-oncogenic roles. Furthermore, galectin-16 is the least-studied member of the family. It has been found to be expressed in different cancer cell lines, and its expression is upregulated by the cAMP activation [155]. Its role in cancer development is still far from understood.

## 3. From Bench to Bedside

### 3.1. Availability of Galectin-Specific Inhibitors

Luckily, due to the small size of galectin proteins, their protein structures and binding site properties have been revealed by numerous nuclear magnetic resonance (NMR) studies [156]. An understanding of protein structures and functions can expedite the drug development process. Galectin inhibitors can be developed in different forms, including (1) small-molecule carbohydrates, (2) natural polysaccharides and their derivatives, (3) peptides and peptidomimetics, and (4) humanised monoclonal antibodies [157]. Their structures and binding affinities to galectin-1, galectin-3, and galectin-9 were fully reviewed by Mariño et al. (2023) [14]. Interestingly, further NMR spectroscopic analyses and competitive ELISA binding assays for eluting protein structure and substrate specifics using synthetic hydroxypropyl methacrylamide (HPMA) copolymers with multi-Galβ4GlcNAc (LacNAc), which is specific to the CRD of human galectin-1 and galectin-3 with differential affinity at the sub-nanomolar level, have been explored [158]. This kind of assays might allow the identification of potential small-molecular inhibitors for future applications. In addition, peptide inhibitors and monoclonal antibodies are also useful to target secreted galectins and not interfere with the intracellular roles of galectins to minimise the side effects. New development techniques for synthesising glycosyl-side chain molecules and amino acid polypeptides may provide further new targets for galectin-based treatment in future.

### 3.2. Applications, Safety/Pitfalls/Limitations, and Ongoing Clinical Trials

At present, all ICIs come in the form of antibodies. This creates a limitation for future treatment potential, especially for brain metastasis, where the antibodies are unable to pass through the blood–brain barrier. In addition, the limited availability of immune cells within the central nervous system also imposes a limitation on the use of ICIs in patients with metastases at sites such as the brain and spine. The development of different small-molecule inhibitors for PD-1/PD-L1 is ongoing [159]. Further understanding of the resident and infiltrating immune cells across to the brain/spine via the blood–brain barrier [160] might make the impossible possible.

T-cell exhaustion is a common phenomenon in treatment using ICIs. Galectin inhibitors are a new approach to improve this situation and the durability of the treatment regimen. Combinations of ICIs and other inhibitors in the form of inhibitory antibodies or TKIs have been the subject of ongoing studies [161]. All the components, including cytotoxic T cells, NK cells, and their inhibitory molecules, within the TME are required to orchestrate the anti-tumorigenic mode of killing tumour cells.

The potential pitfalls of galectin inhibitors in future treatments should be noted in drug development. The first limitation is the unknown factors associated with laboratory settings, such as the differences between mouse models and real humans. The blood components of laboratory mice are different to those of humans. Mice have a high percentage of lymphocytes (around 70% of all leukocytes). In human bodies, the predominant immune cells are neutrophils, which account for over 50% of all leukocytes [162]. This issue is not easily addressed even in humanised mouse models [163]. It is difficult to understand a drug or biological reagent’s effect without human trial studies. In addition, as discussed above, most galectins can alter immune responses. For example, galectin-3 has been shown to play important roles against pathogens [41]. For subjects treated with galectin-3 inhibitors, infection-related adverse effects in subsequent phase 2 or 3 trial studies should be noted as a safety parameter. Treatment using galectin-9 inhibitors also raises similar concerns to galectin-3-inhibitor treatment and can affect different types of immune cells. It may also raise additional concerns regarding autoimmune issues.

Although immune-related adverse events (irAEs) are inevitable in I-O treatment, the occurrence of irAEs during treatment reflects better prognosis and overall outcomes [164]. If the irAEs can be better managed and controlled using corticosteroids, this is a good sign of successful treatment. The effective and safe dosages for combined treatments could be evaluated and determined in clinical trial settings and provide guidelines for regimens and any dosage reductions in the case of irAEs. While clinical trials are still ongoing with many unforeseen hurdles to overcome, galectin-3 small-molecule inhibitors, such as belapectin and GB1211, and galectin-9 humanised monoclonal antibodies still provide new opportunities for combination therapies to tackle the current unmet and unresolved issues associated with ICI treatments for many difficult-to-treat cancers.

## 4. Conclusions

Galectins not only have immunomodulatory roles, but they also contribute to intracellular functions and are expressed in different cancers. Their oncogenic roles can be tackled using small-molecule inhibitors, which can induce cancer cell death in contexts where ordinary ICIs cannot gain access. This is especially important in brain metastasis patients, where only limited immune cells are available for treatment [165]. The ability of antibody delivery and the accessibility of circulating immune cells to tumour sites inside the brain are still largely questionable. Inhibitors, such as those of galectin-3, may provide an alternative option for this kind of patient if the ongoing clinical trials are successful. Combined treatment using galectin inhibitors with TKIs and/or chemotherapy treatment may offer another potential treatment regimen direction. Of course, the related toxicity should be better analysed, and dosage management will be required for different levels of adverse events. Finally, galectins and their inhibitors, as new therapeutic targets, provide more flexibility for future treatment regimens and the addressing of unmet treatment needs.

## Figures and Tables

**Figure 1 biomolecules-13-00902-f001:**
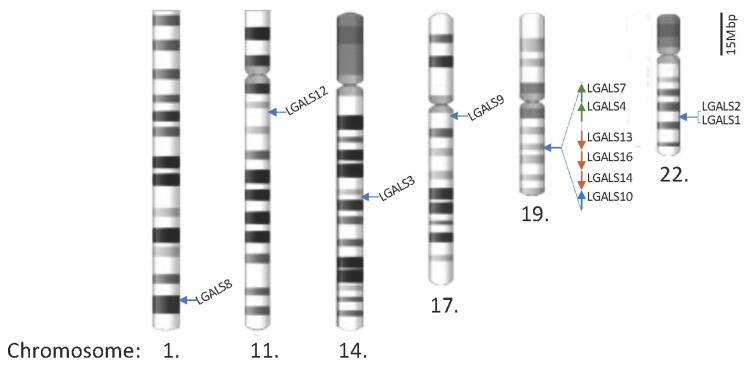
Locations of galectin genes in the human genome. Twenty galectin genes have been identified and are located on chromosomes 1, 11, 14, 17, 19, and 22. The partial q arms of chromosomes 1 and 11 are only shown to reflect the scale, and the scale bar represents 15 megabase pairs (Mbp) in gene distance.

**Figure 2 biomolecules-13-00902-f002:**
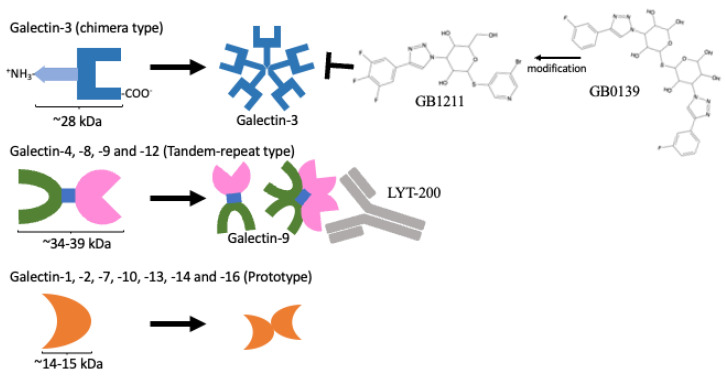
Protein domain structures of galectins and their inhibitors. Only the galectin-3 protein contains CRD domain and an extra amino domain allowing it to form an oligomer (**upper panel**). A protein containing two distinct CRDs, galectin-4, 6, 8, 9, and 12 (**middle panel**). A single CRD protein that can form homo-dimers, galectin-1, 2, 5, 7, 10, 11, 13, 14, and 16 (**lower panel**). GB1211 and LYT-200 represent a newly developed galectin-3-specific inhibitor [15] and a humanised monoclonal antibody against galectin-9, respectively, and are currently under clinical trial.

**Figure 3 biomolecules-13-00902-f003:**
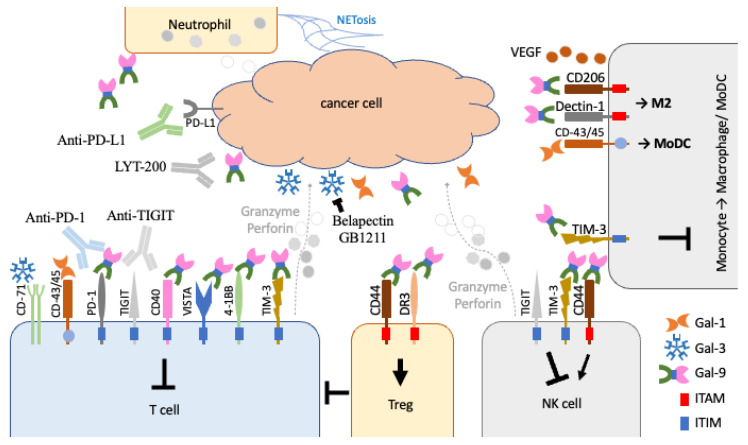
Effects of galectin-1, -3, and -9 in the control of different types of immune cells within the TME. Galectin-1 (Gal-1) and -3 (Gal-3) have been shown to bind to CD45 and CD-71 to inhibit effector T cells (T cell). In contrast, galectin-9 (Gal-9) binds to TIM-3 and numerous ITIM-containing immunomodulators on effector T cells and NK cells for tumoural immunosuppression. It can also act on Tregs through CD44 and DR3 to inhibit effector T cells. It can also promote M2 macrophage polarisation and VEGF secretion in angiogenesis. The recruited neutrophils may also contribute to galectin-9 secretion within the environment.

**Table 1 biomolecules-13-00902-t001:** Clinical trial studies (extracted from ClinicalTrials.gov, accessed on 11 May 2023).

Target	Drug	Phase	Cancer Type	Intervention
Galectin-1	OTX008	I	Solid tumours	NCT01724320 Status unknown (updated: 2012)
Galectin-3	Belapectin (GR-MD-02)	I	Metastatic melanoma	NCT02117362 Completed (updated: 2019)
		I	Metastatic melanoma, NSCLC, HNSCC	NCT02575404 [21] Active (updated: 2022)
	GB1211	I	Healthy subjects	NCT03809052 [16] Completed (updated: 2021)
		I/II	NSCLC	NCT05240131 Recruiting (updated: 2023)
	GCS-100	I/II	Relapsed/Refractory diffuse large-B-cell lymphoma	NCT00776802 Withdrawn as funding issue (updated: 2013)
	GM-CT-01	I	Breast, colorectal, head and neck, lung, prostate	NCT00054977 Completed (updated: 2012)
	PectaSol-C, modified citrus pectin (MCP) [22]	N/A	Non-cancer-related: study for high blood pressure control	NCT01960946 Completed (updated: 2021)
	Galactomannan/ ProLectin-M [23]	III	Non-cancer-related: antagonist for COVID-19	NCT05096052 Recruiting (updated: 2022)
Galectin-9	LYT-200 (monoclonal antibody against galectin-9)	I	Acute myeloid leukaemia	NCT05829226 Recruiting (updated: 2023)
		I/II	Metastatic cancer in head and neck, colorectal, pancreatic, or urothelial origins	NCT04666688 Recruiting (updated: 2023)

**Table 3 biomolecules-13-00902-t003:** Expression in different cancers [100,101,102,103,104].

Cancer Type	Galectin-1	Galectin-2	Galectin-3	Galectin-4	Galectin-7	Galectin-8	Galectin-9	Galectin-12	Galectin-13	Galectin-14	Galectin-16
Melanoma	+		+		+		+				
Lungs	+		+	+		+	+		+	+	+
Glioma/ Neuroblastoma	+		+	+	+	+	+				
Leukaemia			+					+			
Myeloma							+				
Lymphoma	+		+		+						
Thyroid	+		+		+				-	-	-
Oral			+			+	+		+		
Gastric	+		+		+						
Liver	+	!	+	+		+	+				
Pancreas	+		+	+			+				
Colorectal	+	+	+	+	+	+	+	+			
Renal			+		+	+	+				
Bladder	+		+		+	+					
Breast	+	+	+	+	+	+	+		+	+	+
Cervical	+		+	+	+		+	+			
Endometrial									+		
Ovarian	+		+	+	+	+	+			+	
Prostate	+		+			+	+				
Testis									+	+	+

Key: +—oncogenic; -—tumour-suppressive; !—controversial.

## Data Availability

Not applicable.

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
