# Peer review of "Chimera and Tandem-Repeat Type Galectins: The New Targets for Cancer Immunotherapy"

_biomolecules, 2023, doi:10.3390/biom13060902_

Round 1

Reviewer 1 Report

Authors have presented an interesting manuscript on emerging topic related to the association of galectins with cancer immunotherapy, their cytological properties and linkage of various galectins with each subtypes. The article is original, well structured; easy to read with main emphasis on the detailed analysis of different types of galectins as biomarkers and their immunosuppressive and oncologic functions for anticancer. Authors have described the concept to a greater extent but the manuscript still need some major corrections before publishing.

·       First major concern about this manuscript is title not properly framed. I am unable to understand the term “more” in title of manuscript, authors are suggested to reframe the title asper the suggestions. Authors are suggested to highlight the novelty of this review article as there are number of review articles are available on the same topic.

·       Abstract must be improved in terms of addition of more content that could be better correlated with the manuscript.

·       Manuscript is not properly formatted as per the journal’s guidelines, at some places headings were represented as numbering system while subsections were  not properly numbered.

·       Figure 2 should be formatted properly.

·       Authors should add more content in section summary from bed to bench side.

·       Authors are suggested to include latest studies related to the theme:

https://doi.org/10.3389%2Ffimmu.2022.1104625

https://doi.org/10.3390/ph15030335

·       Authors are suggested to include conclusion part with future directions.

·       Abbreviations should be clearly stated throughout the manuscript.

Authors are suggested to proofread the manuscript for grammatical and language with a native English speaker. Rectify the grammatical errors.

Author Response

Dear Reviewer,

Thanks for your comments and suggestions on our manuscript. We have made the amended with your comments and the structure of our manuscript. Please kindly review and see any additional amendment needed.

Best regards,

Frankie Ko

Department of Medicine,

The University of Hong Kong

102 Pokfulam Road, Queen Mary Hospital

Hong Kong

Addressing on reviewer 1 comments:

  1. Title issue …more”. Authors are suggested to reframe the title as the per suggestions. Highlight the novelty of this review article.

Response: The title has been changed in amended manuscript as “Chimera and Tandem-Repeat Type Galectins: The New Targets for Cancer Immunotherapy”.

  1. Abstract must be improved in terms of addition of more content that could be better correlated with the manuscript.

Response: The abstract has been rewritten and should be better correlated with the content of manuscript.

  1. Manuscript is not properly formatted as per the journal’s guidelines, at some places headings were represented as numbering system while subsections were not properly numbered.

Response: The format style of the amended manuscript has been updated according to Biomolecules (MDPI) guidelines.

  1. Figure 2 should be formatted properly.

Response: The Figure 2 and 3 of the amended manuscript has been updated according to Biomolecules (MDPI) guidelines.

  1. Authors should add more content in section summary from bed to bench side.

Response: “The summary: from bench to bed side” has been extended and rearrange to provide better clarification and discussion on the translational view in newly amended manuscript section 3: From Bench to Bedside.  

  1. Authors are suggested to include latest studies related to the theme:
  • [Ref 6]: Pandey P, Khan F, Qari HA, Upadhyay TK, Alkhateeb AF and Oves M. Revolutionization in Cancer Therapeutics via Targeting Major Immune Checkpoints PD-1, PD-L1 and CTLA-4. Pharmaceuticals 2022, 15, 335.
  • [Ref 155]: Laderach DJ and Compagno D (2023) Inhibition of galectins in cancer: Biological challenges for their clinical application. Front. Immunol. 13:1104625.

Response: Thanks for the suggestions. These 2 latest studies have been incorporated into the amended manuscript in part 1 Introduction and part 3.1 Availability of galectin-specific inhibitors respectively.

  1. Authors are suggested to include conclusion part with future directions.

Response: The section “4. Conclusion” has been also added in amended manuscripts.

  1. Abbreviations should be clearly stated throughout the manuscript.

Response: An “Abbreviations” part has been added at the end of the manuscript before References part. All abbreviations have been checked with consistence. 

  1. Authors are suggested to proofread the manuscript for grammatical and language with a native English speaker. Rectify the grammatical errors.

Response: Thank you for the suggestion. I know that my writing skills are not good enough and the manuscript has been further edited by English-edited service and my co-authors.

Reviewer 2 Report

Galectins: The novel targets for cancer immunotherapies and more

Biomolecules

This review manuscript aims to highlight the role of galectins in cancers. However, there are major concerns that require extensive revision.

The reader can easily identify numerous mistakes, usage of unusual and unacceptable terms, long sentences and etc. The abstract requires extensive editing.

The introduction should be starting on galectins not immune checkpoint inhibitors. There is no clear evidence that galectins interfere with every single ICIs per the author’s introductory section. Although Gal-9 interacts with TIM-3 and PD-1, these studies were performed in mice. There is no clinical evidence about the interference of galectins with ICIs in human studies, therefore, the authors should not make generalized conclusions.

There is a study that has shown a higher Gal-9 mRNA level in non-responders to Avelumab but has not been discussed.

Galectins are not only present in the cytosol but also present on the cell surface and in the nuclei.

Galectins have different receptors and the potential receptors for each galectin should be discussed.

The authors have not done a thorough literature review to discuss the vast knowledge of Galectins.

For example, Gal-9 interacts with different receptors and the authors should acknowledge that galectins have immunomodulatory functions. They may exert stimulatory or inhibitory functions.

Gal-9 interacts with CD44, CD137, PDI, PD-1, TIM-3, etc. Therefore, when gal-9 interacts with CD44 induces T cell, and NK cell activation but via interaction with TIM-3 results in T cell exhaustion.

Gal-9 is the most studied human galectin and there are extensive reports on the immunomodulatory role of this lectin in different contents. Unfortunately, the authors have not discussed such studies. For instance, Gal-9 is expressed on Tregs and via interaction with TIM-3 impairs T cell effector functions (PMCID: PMC3324980).

Gal-9 is reported to be an exhaustion marker in HIV and cancer but are not discussed here.

The role of Gal-9 in NK cell functions should be discussed and cited. Gal-9 interaction with CD44 on NK cells results in NK cell activation.  

A recent study has reported that Gal-9 depletes MAIT cells in CLL. The elevation levels of Gal-9 in CLL and its impact on MAIT cells should be discussed.

The potential source of galectins should be discussed for example neutrophils are reported to be the source of Gal-9.

Galectins can be detected as truncated and full lengths and their functions are not the same. For example, truncated Gal-8 exhibits much lower stimulatory properties than the full length (https://pubs.acs.org/doi/full/10.1021/acscentsci.3c00054).

In Figure 3, the authors have shown that Gal-9 interacts with TIGIT and TIM-3 on NK cells. However, they have missed the information that Gal-9 also interacts with CD44 on NK cells.  

·         In the same figure, the authors have claimed that Gal-9 interacts with different molecules in T cells. However, they have missed protein disulfide isomerase (PDI), and CD44. I am not sure why authors have suggested that Gal-9 interacts with TIGIT. Moreover, Gal-9 interacts with TCR and should be included and discussed.

Galectins are highly abundant in the plasma, extracellularly and intracellularly in all immune and non-immune cells. Therefore, inhibiting Galectins should not be considered as a straightforward approach.

The abundance of Galectins indicates their diverse biological properties in health and disease. As such any therapeutic approach for blocking or inhibiting galectins might be impractical or result in diverse immunological adverse reactions.

In line 169/170, the authors have mentioned that gal-9 is expressed on different immune cells such as T cells, NK, Tregs, macrophages, and monocytes. They should include other cells such as neutrophils as the most abundant immune cells in humans.

The authors appear to conclude that Gal-9 is mainly an immunosuppressive protein but this is not the case (lines 180-181). Gal-9 depending on the interacting receptor can be either stimulatory or inhibitory. Is has been shown that Gal-9 is associated with T cell activation via interaction with CD44 and TCR.

Once again, this manuscript suffers from a poor English language and seriously requires an editing service.

Due to the poor language, the structure of some sentences is vague and hard to understand.

The limitations and pitfalls of targeting galectins have been discussed but require editing and clarification.  

The quality of language is extremely poor. 

Author Response

Dear Reviewer,

Thanks for your substantial comments and insights especially on galectin-9. We made the amended with your comments and answered your queries as following points. Please kindly review and see any additional amendment needed.

Best regards,

Frankie Ko

Department of Medicine,

The University of Hong Kong

102 Pokfulam Road, Queen Mary Hospital

Hong Kong

Addressing on reviewer 2 comments (with rearrangement on questions’ order):

  1. Numerous mistakes, usage of unusual and unacceptable terms, long sentences and etc. The abstract requires extensive editing.

Response: The manuscript has been further edited by co-authors and English-editing service. I feel deep sorry on the careless grammatical mistakes.

  1. The introduction should be starting on galectins not immune checkpoint inhibitors. The interferences between galectins and ICIs have no clear evidence and making generalized conclusions.

Response: In review, we would like to emphasize that galectins are the new targets for immunotherapeutic treatment. We introduced ICIs for providing background on current clinical setting on immunotherapy. We agree that the trial is ongoing and result is not yet completely released on galectin-9 inhibitor LYT-200 (NCT04666688). Three arms of this study, (1) one of the arm sole treatment on metastatic solid tumour patients and (2) combined treatment with current ICI/Tislelizumab or (3) Gemzar/nab-paclitaxel with provide more information on the efficacy of galectin-9 inhibitor in near future. We just would like provide information on most recent trial study but not to make a generalized conclusion in the Introduction part of last manuscript. 

  1. Galectins are not only present in the cytosol but also present on the cell surface and in the nuclei.

Response: Agree to reviewer comment, galectins not only in cytoplasmic and cell surface but also inside nucleus. Nuclear interacting partners are also included in Table 2 and the corresponding galectin sections 2.1-2.3.

  1. Galectins have different receptors and the potential receptors for each galectin should be discussed.

Response: Galectins have numerous receptors and interacting partners. In this review, we would like to focus their immunological roles to fit our aim to address the potentials of using galectins and their inhibitors for immune-oncological treatments. Key partners are listed in Table 2 to fit our purpose.

  1. The authors have not done a thorough literature review to discuss the vast knowledge of galectins.

Response: As mentioned above, we would like to focus on their immunological roles. We know that galectins involving very broad biological functions.

  1. There is a study that has shown a higher Gal-9 mRNA level in non-responders to Avelumab but has not been discussed.

Response: Higher Gal-9 mRNA level in tumour infiltrating lymphocytes (TILs) as exhaustion marker to Avelumab treatment has been included in section 2.2.1. Galectin-9… , first paragraph last 4 lines with [reference 98] Okoye, I et al. (2020) J Immunother Cancer 8(2), e001849 added.

  1. Gal-9 interacts with different receptors and the authors should acknowledge that galectins have immunomodulatory functions. They may exert stimulatory or inhibitory functions.

Response: The immunomodulatory functions has emphasized in Abstract and incorporated into the amended section 2.2.1. Galectin-9…., second paragraph, 4th line.

  1. Gal-9 interacts with CD44, CD137, PDI, PD-1, TIM-3, etc. Hence, when gal-9 interacts with CD44 induces T cell, and NK cell activation but via interaction with TIM-3 results in T cell exhaustion.

Response: Agree, The Gal-9 acts on TIM-3 and has similar roles to TIGIT for T cell exhaustion role and discussed in section 2.2.1. Galectin-9…., second paragraph.

  1. Gal-9 is the most studied human galectin and there are extensive reports on the immunomodulatory role of this lectin in different contents. Unfortunately, the authors have not discussed such studies. For instance, Gal-9 is expressed on Tregs and via interaction with TIM-3 impairs T cell effector functions. (PMCID: PMC3324980- Elahi S, Dinges W, Lejarcegui N et al. (2011) Protective HIV-specific CD8+ T cells evade Treg cell suppression. Nat Med 17, 989–995.).

Response: Thanks for the suggestion, Gal-9-TIM-3 axis for Treg suppressing cytotoxic T CD8+ cells has been discussed in section 2.2.1. Galectin-9… second paragraph with suggested reference [107] added.

  1. Gal-9 is reported to be an exhaustion marker in HIV and cancer but are not discussed here.

Response: Thanks for the comment. It is important to state that Gal-9 is an exhaustion marker and has been discussed and incorporated into first paragraph of section 2.2.1. Galectin 9… last 3rd line.

  1. The role of Gal-9 in NK cell functions should be discussed and cited. Gal-9 interaction with CD44 on NK cells results in NK cell activation.

Response: The CD44 in NK cell activation has been discussed in third paragraph of section 2.2.1. Galectin-9…, third paragraph of amended manuscript.

  1. A recent study has reported that Gal-9 depletes MAIT cells in CLL. The elevation levels of Gal-9 in CLL and its impact on MAIT cells should be discussed.

Response: The discussion of MAIT has been incorporated into last paragraph of section 2.2.1. Galectin 9…. on amended manuscript.

  1. Lines 169-170: The authors have mentioned that gal-9 is expressed on different immune cells such as T cells, NK, Tregs, macrophages and monocytes. They should include other cells such as neutrophils as the most abundant immune cells in humans.

Response: Agree with reviewer, neutrophil is the most abundant leukocytes in humans. The neutrophil component is added in Figure 3 and discussed in last secondparagraph of section 2.2.1. Galectin-9…. On clincal side, neutrophil infiltration to tumour site is marked as poor prognosis in solid cancer patients. The neutrophil role is also discussed in this paragraph.

  1. The potential source of galectins should be discussed for example neutrophils are reported to be the source of Gal-9.

Response: ,Neutrophils can produce gal-9 in TME. However, as mentioned above neutrophil infiltration is correlated to poor prognosis. Gal-9 in TME may play pro-tumourgenic role and discussed in last second paragraph of section 2.2.1 Galectin-9….

  1. In figure 3, the authors have shown that Gal-9 interacts with TIGIT and TIM-3 on NK cells. However, they have missed the information that Gal-9 also interacts with CD44 on NK cells.

Response: Agree. The TIGIT and Gal-9 interaction in Figure 3 is misled and amendment on Figure 3 is made. The discussion on Gal-9-CD44 activation has also been discussed in third paragraph of section 2.2.1. Galectin-9…..

  1. In the same figure, the authors have claimed that Gal-9 interacts with different molecules in T cells. However, they have missed protein disulfide isomerase (PDI), and CD44. I am not sure why authors have suggested Gal-9 interacts with TIGIT. Moreover, Gal-9 interacts with TCR and should be included and discussed.

Ans: The PDI and TCR included in “Extracellular” column of Table 2 and the interaction with TCR is also discussed in second paragraph line 13-14 of section 2.2.1. Galectin-9… and Figure 3 has amended on this figure.

  1. Lines 180-181: The authors appear to conclude that Gal-9 is mainly an immunosuppressive protein but this is not the case. Gal-9 depending on the interacting receptor can be either stimulatory or inhibitory. Is has been shown that Gal-9 is associated with T cell activation via interaction with CD44 and TCR.

Response: In third paragraph of section 2.2.1. galectin-9…, the interaction  of CD44, TCR and  Gal-9 has been discussed. In addition, galectin-9 could induce cytotoxic human T cell apoptosis in T cell/tumour cell co-culture. The role of galectin-9 within TME may serve as protumourgenic role.

  1. Galectins can be detected as truncated and full lengths and their functions are not the same. For example, truncated Gal-8 exhibits much lower stimulatory properties than the full length. Chen SA, Arutyunova E, Lu J et al. (2023) ACS Cent Sci, 9(4), 696–708.

Response: Agree. Numerous isoforms have been detected in different proteins such as galectin-8. Currently galectin-8 role in cancer is still unknown and the mentioned reference is more related to anti-viral properties on gal-8 in COVID inhibition. It is not the primary aim of this review.

  1. Galectins are highly abundant in the plasma, extracellularly and intracellularly in all immune and non-immune cells. Therefore, inhibiting Galectins should not be considered as straightforward approach.

Response: Agree. For example, the galectin-1 have too diversity role in cell functions and should not be a good candidate and has been discussed in Section 2.3.1. Galectin 1… last paragraph. Combination treatment for preventing T cell exhaustion will be the idea of using galectin-9 inhibitor to orchestrate anti-tumour effect within the TME as discussed in the second paragraph of Section 3.2.

  1. The abundance of galectins indicates their diverse biological properties in health and disease. As such any therapeutic approach for blocking or inhibiting galectins might be impractical or result in diverse immunological adverse reactions.

Response: The use of the inhibitors can be as antibody mentioned in fourth paragraph of section 2.2.1 Galectin-9… to limit the harmful effect on the intracellular roles of galectin-9. Animal testings and human trials are required to address in toxicity of the inhibitors, which has been mention in section 3.2.

  1. The limitations and pitfalls of targeting galectins have been discussed but require editing and clarification.

Response: The section 3 has been amended and edited.

  1. Extensive editing of English language must be required.

Response: I understand my writing skills are not good enough. My manuscript has been further edited English-editing service and reviewed by co-authors to make it more readable.

Round 2

Reviewer 1 Report

Authors have substantially revised the manuscript as per the suggestions, therefore the manuscript can now be considered for publication.

Moderate English and grammatical changes are required.

Author Response

Dear Reviewer,

Thanks for your comment on our manuscript. The manuscript has been further amended for clear presentation.

Best regards,

Frankie Ko

Department of Medicine,

The University of Hong Kong

102 Pokfulam Road, Queen Mary Hospital

Hong Kong

Reviewer 2 Report

The manuscript has substantially improved. However, the authors should make sure to cite relevant references throughout the test. For example, I have noticed statements without proper referencing.  Page 9 line 7, the reference for this statement is missing " neutrophils also contribute to the secretion of galactin-9"

Author Response

Dear Reviewer,

Thanks for your comment on our manuscript.

The missing citation has been added as [117] in page 9 Line 7. The manuscript has also been checked and with some minor amendments to correct the errors. 

Best regards,

Frankie Ko

Department of Medicine,

The University of Hong Kong

102 Pokfulam Road, Queen Mary Hospital

Hong Kong